# Vision-Language-Action in Robotics: A Survey of Datasets, Benchmarks, and Data Engines

**Ziyao Wang[1], Bingying Wang[2], Hanrong Zhang[3], Tingting Du[4], Tianyang Chen[1], Guoheng Sun[1], Yexiao He[1], Zheyu Shen[1], Wanghao Ye[1], Ang Li[1]**

## Abstract

Despite remarkable progress in Vision–Language–Action (VLA) models, a central bottleneck remains underexamined: the data infrastructure that underlies embodied learning. In this survey, we argue that future advances in VLA will depend less on model architecture and more on the co-design of high-fidelity data engines and structured evaluation protocols. To this end, we present a systematic, data-centric analysis of VLA research organized around three pillars: datasets, benchmarks, and data engines. For datasets, we categorize real-world and synthetic corpora along embodiment diversity, modality composition, and action space formulation, revealing a persistent fidelity-cost trade-off that fundamentally constrains large-scale collection. For benchmarks, we analyze task complexity and environment structure jointly, exposing structural gaps in compositional generalization and long-horizon reasoning evaluation that existing protocols fail to address. For data engines, we examine simulation-based, video-reconstruction, and automated task-generation paradigms, identifying their shared limitations in physical grounding and sim-to-real transfer. Synthesizing these analyses, we distill four open challenges: representation alignment, multimodal supervision, reasoning assessment, and scalable data generation. Addressing them, we argue, requires treating data infrastructure as a first-class research problem rather than a background concern. To support the community, we release a continuously updated repository at `https://github.com/ziyaow1010/vla-datasets-benchmarks`.

## 1 Introduction

Building robots that can follow natural language instructions across diverse tasks and environments is a central goal of embodied AI. Vision–Language–Action (VLA) models pursue this goal by learning to map visual observations and language instructions directly to actions (Kim et al., 2024; Ma et al., 2024). Unlike conventional manipulation systems designed for fixed tasks in controlled settings, VLA models are expected to generalize across objects, environments, and task variations (Kawaharazuka et al., 2025). This flexibility is essential for robots operating in open-ended or human-centered environments, where the full range of situations cannot be specified in advance.

VLA has attracted growing interest, driven by progress across several fronts. Large vision–language models (VLMs) have improved substantially in visual perception and instruction following (Zhang et al., 2024a; Dai et al., 2023), providing a strong backbone for grounding language in visual context. The growth of robot demonstration datasets and advances in imitation learning have also made it more practical to train action policies directly from human demonstrations (Shao et al., 2025; Zhang et al., 2025a; Xiang et al., 2025). Improvements in simulation tools, sim-to-real transfer methods, and edge computing hardware have further narrowed the gap between research prototypes and real deployments (Todorov et al., 2012; Zhu et al., 2020; Sun et al., 2026). Together, these advances have shifted VLA from a largely theoretical direction toward systems that can interact with the physical world and generalize across a wide range of tasks.

Despite this progress, building reliable VLA systems remains difficult, and the core challenge is increasingly about data and evaluation rather than model design alone. On the data side, VLA datasets vary widely in

---

[1]University of Maryland, College Park   [2]University of Utah   [3]Northeastern University   [4]University of Wisconsin–Madison.   Correspondence to: Ang Li <angliece@umd.edu>

modality coverage, action representations, robot platforms, task definitions, and instruction styles (Padalkar et al., 2023; Liu et al., 2023). Real-world demonstrations are expensive to collect and typically cover only a narrow range of tasks and environments (Khazatsky et al., 2024). Synthetic and simulation data can be produced at scale, but often fail to reflect real-world conditions closely enough, causing models to break down when the scene, robot, or task changes (Deng et al., 2025). On the evaluation side, the field lacks standardized protocols: different works use different task definitions, success criteria, and data splits, which makes it difficult to compare methods or determine whether reported improvements reflect genuine generalization. Therefore, a systematic study and categorization of VLA-related datasets, benchmarks, and data engines for scalable data production is an urgent need to better understand current limitations and support more reliable progress in the field.

In this survey, we address this gap by providing a structured and data-centric analysis of VLA research. We organize existing work into three primary categories: **datasets**, **benchmarks**, and **data engines**. Datasets refer to curated collections of demonstrations used for training VLA models; benchmarks define standardized evaluation protocols, task settings, and metrics for evaluating performance and generalization; data engines are scalable systems or pipelines designed to collect, generate, or augment VLA training data. Although these categories may overlap in practice, the three-way organization allows us to examine each component clearly and compare design choices in a systematic way. Within each category, we further introduce finer-grained subgroups: datasets are divided into *synthetic* datasets and *real-world* datasets; benchmarks are organized by *task settings* (e.g., tabletop versus non-tabletop), *episode horizon*, and *task complexity*; and data engines are classified into *video-to-data pipelines*, *hardware-assisted data collection systems*, and *generative data engines*. For each subgroup, we summarize representative works, analyze their design choices, and discuss their strengths and limitations. We also identify open challenges and outline promising directions for future VLA data and evaluation research. For the convenience of locating and compare relevant works, we provide a tree-structured taxonomy that summarizes representative VLA data-related works from 2023 to 2025 and clarifies their relationships within a unified framework in Figure 1. To the best of our knowledge, this is the first survey to systematically analyze the VLA field from a data-centric perspective.

Our contributions are summarized as follows:

- We present a unified data-centric taxonomy for VLA research that organizes existing works into datasets, benchmarks, and data engines, clarifying their distinct roles in training, evaluation, and scalable data production.
- We introduce a structured analytical framework for understanding VLA data resources, including a two-dimensional characterization of benchmarks and a systematic comparison of dataset and data engine design choices.
- We identify three structural challenges that cut across all three components: representation alignment across embodiments, reliable evaluation of long-horizon compositional tasks, and scalable data generation that preserves physical realism. For each, we outline concrete directions for future work.

## 2 Preliminaries

This survey focuses on datasets, benchmarks, and data engines for VLA learning in robotic manipulation. While the VLA paradigm also applies to other embodied domains such as autonomous driving or mobile navigation, we restrict our scope to robotic systems where actions correspond to controlling robot arms and, optionally, grippers. Accordingly, all datasets and benchmarks discussed in this paper study vision- and language-conditioned manipulation rather than navigation-only or driving scenarios. Figure 2 shows the scope of the survey.

We formulate the VLA problem as a sequential decision-making process in which a policy $\pi$ maps visual observations and natural language instructions to robot control actions. At each time step $t$, the policy receives a visual observation $o_t$ together

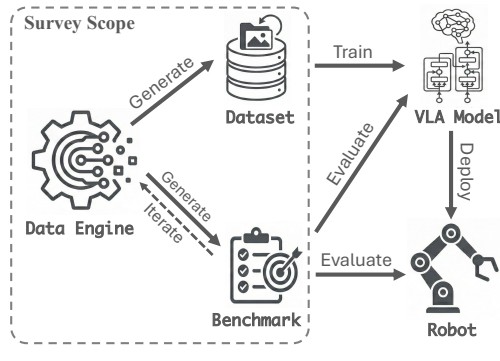

Figure 2: Scope of this survey.

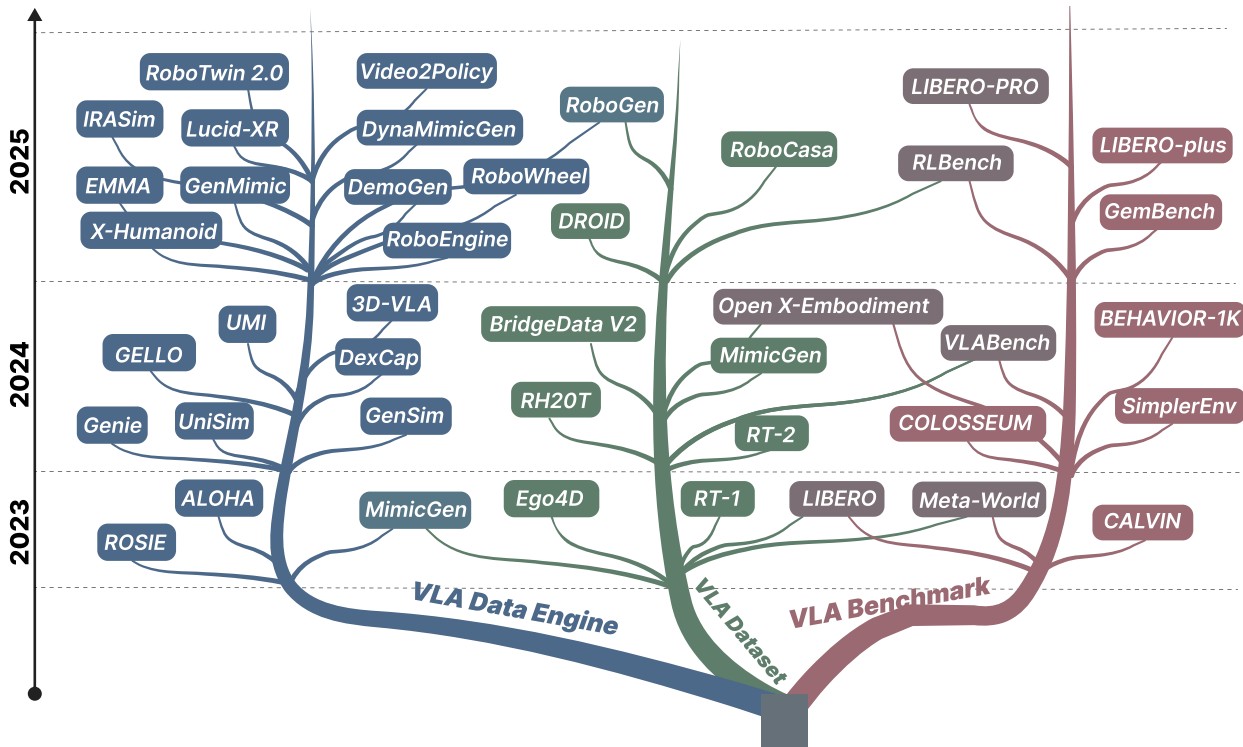

Figure 1: Taxonomy and Temporal Landscape of VLA Data-Centric Works (2023-2025).

with a language instruction $l$, and outputs an action $a_t$, written as $a_t = \pi(o_t, l)$. The visual observation $o_t$ may be represented as a single RGB image, 3D RDB-D representation such as a point cloud, or a video sequence capturing temporal dynamics. The language instruction $l$ typically specifies a high-level task goal and remains fixed over an episode.

The action $a_t$ corresponds to a robot control command, and existing *VLA datasets* mainly differ in their action representations along two axes: the control target and the parameterization. In terms of control target, actions are defined either in end-effector (EEF) space or in joint (DoF) space. EEF actions specify the desired motion of the robot end-effector in task space, typically including Cartesian position and orientation, whereas DoF actions directly command individual joint states of the robot. In terms of parameterization, actions can be represented as either absolute targets or relative (delta) commands. An absolute action specifies a target state in the chosen action space, while a delta action specifies an incremental change with respect to the current state, formally defined as $a_t^{\Delta} = a_t^{\mathrm{abs}} - a_{t-1}^{\mathrm{abs}}$. These design choices define the structure of the action space provided by a dataset and serve as key dimensions for dataset categorization in Section 3.

*VLA benchmarks* evaluate a trained policy on a collection of tasks under predefined environments and protocols. Benchmarks primarily vary in the structure of the task and the structure of the environment. Tasks range from short-horizon atomic manipulations to long-horizon, compositional instruction following, while environments may consist of constrained tabletop settings or more diverse multi-scene configurations. Performance is most commonly measured using task success rate. Given a set of evaluation episodes $\mathcal{E}$, the success rate is defined as $\mathrm{SR} = \frac{1}{|\mathcal{E}|} \sum_{e \in \mathcal{E}} \mathbb{I}[\text{task completed in } e]$, where $\mathbb{I}[\cdot]$ is an indicator function. Although alternative metrics such as progress-based scores are sometimes adopted, success rate remains the dominant evaluation criterion across existing VLA benchmarks. These definitions establish a unified vocabulary for analyzing datasets and benchmarks in the following sections.

In addition to datasets and benchmarks, we introduce the notion of a *data engine* for VLA learning. A data engine refers to a system or pipeline that continuously generates, transforms, or augments training data, rather than providing a fixed, static dataset. Formally, a data engine can be viewed as a data generation process $\mathcal{G}$ that produces task instances, trajectories, or environment variations conditioned on available inputs

such as videos, simulation states, human demonstrations, or language prompts. Unlike conventional datasets that are collected once and reused, data engines are designed to scale data diversity over time and to adapt to new embodiments, tasks, or environments. In this survey, we categorize VLA data engines into three main paradigms: video-to-data engines, which convert human videos into robot-executable supervision; hardware-assisted engines, which collect demonstrations through embodied teleoperation or sensing interfaces; and generative data engines, which synthesize tasks, trajectories, or environments using simulation and generative models. These definitions complete the conceptual framework used throughout the remainder of the paper.

## 3 VLA Datasets

Datasets serve as the foundational learning material for VLA models and strongly influence how agents generalize across embodiments and environments. Following our taxonomy, we categorize existing datasets primarily by their **data source**: *Real-World Datasets* (§3.1) and *Synthetic Datasets* (§3.2). Within this categorization, we analyze differences in **embodiment diversity**, **action representation**, and **modalities**, as summarized in Table 1.

### 3.1 Real-World Datasets

Real-world datasets are collected from physical robot operations and include high-fidelity interaction data that reflects true contact dynamics and friction. These datasets provide real images, authentic robot states and actions, and physically grounded signals that remain difficult to reproduce accurately in simulation environments. As a result, real-world data is generally regarded as high quality but also high cost. It is inherently difficult to scale due to substantial human labor requirements (data collectors must operate in physical environments) and infrastructure expenses (real robots, physical scenes, and mature manipulation systems are required). Table 1 displays a few representative real-world datasets for VLA training.

Open X-Embodiment (Padalkar et al., 2023) is one of the most widely used pretraining datasets for VLA models, as it aggregates data across diverse robot platforms and institutions. Its large scale and cross-platform coverage make it particularly suitable for pretraining, enabling models to acquire general visual grounding and action capabilities. At the same time, the heterogeneity in action interfaces and control frequencies introduces valuable diversity, facilitating adaptation to specific robots and downstream manipulation tasks. In contrast, single-embodiment baselines such as RT-1 (Brohan et al., 2022) and BridgeData V2 (Walke et al., 2023) emphasize data hygiene and consistency. These datasets are collected on relatively fixed robot platforms, and are therefore more specific in embodiment. They are particularly suitable for fine-tuning models when deployment targets the same or similar robot systems.

Beyond embodiment scale, task diversity and modality design are also critical factors in dataset construction for VLA. Different datasets emphasize different aspects of task and scene design to improve generalization and robustness. For example, DROID (Khazatsky et al., 2024) increases visual and environmental variation (e.g., backgrounds and lighting) to enhance perceptual robustness under real-world conditions. Multimodal datasets such as RH20T (Fang et al., 2023) further incorporate tactile/force and audio signals, which are particularly beneficial for contact-rich manipulation where vision alone may be insufficient. In addition to collecting robot demonstrations, recent work explores leveraging large-scale human hand–object interaction (HOI) data as complementary supervision. Approaches such as RT-2-style co-training (Brohan et al., 2023) and human video corpora like Ego4D (Grauman et al., 2022) connect robot learning with web-scale or egocentric human data to introduce broader semantic priors.

Datasets should be selected according to the desired generalization objective and deployment setting. For cross-embodiment pretraining and large-scale transfer, aggregated corpora such as Open X-Embodiment (O'Neill et al., 2025) are well suited, as their scale and platform diversity support the learning of transferable representations across heterogeneous robot interfaces. When the target deployment involves a specific robot platform, single-embodiment datasets such as RT-1 (Brohan et al., 2022), DROID (Khazatsky et al., 2024), or BridgeData V2 (Walke et al., 2023) are often more appropriate for fine-tuning, since their controlled settings and consistent action interfaces better match downstream execution conditions. If robustness to environmental variation is required, distributed real-world collections such as DROID (Khaz-

Table 1: Overview of representative VLA datasets. We summarize each dataset by embodiment, modality composition, and action space formulation.

| Dataset | Embodiment | Img | Vid | Txt | D | Action | Key Feature |
|---|---|---|---|---|---|---|---|
| Real-World Datasets | | | | | | | |
| **Open X-Embodiment** | 22 robots | ✓ | ✗ | ✓ | ✓ | Mixed EEF | Cross-embodiment aggregation |
| **RT-1** | Everyday Robots | ✓ | ✗ | ✓ | ✗ | Delta EEF | Fleet-scale teleoperation |
| **RT-2** | Everyday Robots | ✓ | ✗ | ✓ | ✗ | Delta EEF | Web co-training |
| **DROID** | Franka Panda | ✓ | ✗ | ✓ | ✓ | Absolute EEF | In-the-wild collection |
| **BridgeData V2** | WidowX 250 | ✓ | ✗ | ✓ | ✓ | Delta EEF | Low-cost standardized setup |
| **RH20T** | 4 robots | ✓ | ✓ | ✓ | ✓ | EEF+DoF | Multimodal contact data |
| **Ego4D** | Human hands | ✗ | ✓ | ✓ | ✗ | N/A | Human interaction corpus |
| Synthetic Datasets | | | | | | | |
| **SynGrasp-1B** | Franka Panda | ✓ | ✗ | ✓ | ✗ | Delta EEF | Large-scale grasp synthesis |
| **RoboCasa** | Franka Panda | ✓ | ✗ | ✓ | ✗ | EEF | Kitchen simulation suite |
| **RoboGen** | Franka/Legged robot | ✓ | ✗ | ✓ | ✗ | EEF+DoF | LLM-generated tasks |
| **MimicGen** | 4 robots | ✓ | ✗ | ✗ | ✗ | Delta EEF | Demonstration augmentation |

atsky et al., 2024) provide diverse lighting and scene configurations. For contact-rich manipulation tasks, multimodal datasets such as RH20T (Fang et al., 2023) offer additional tactile or force signals that complement vision. Finally, when transferring semantic knowledge from large-scale human data is beneficial, web-co-trained or human-centric sources such as RT-2 (Brohan et al., 2023) and Ego4D (Grauman et al., 2022) can introduce broader semantic priors, although embodiment differences and action-label mismatches must be carefully addressed.

Despite their realism, existing real-world datasets do not fundamentally resolve the quality–cost trade-off, as large-scale physical data collection remains expensive and difficult to scale. Achieving low-cost acquisition of high-quality data therefore remains a central challenge for future VLA research.

## 3.2 Synthetic Datasets

Considering the high cost and limited scalability of real-world datasets, researchers often address data scarcity by generating synthetic data in simulators (Todorov et al., 2012). Simulation environments allow explicit specification of scenes and tasks, and provide built-in tools for sampling grasp poses, solving inverse kinematics, and automatically determining task success or failure. As a result, synthetic datasets can be efficiently scaled by increasing the number of trajectories per task or varying the configuration of the scene. However, the fidelity of synthetic data is fundamentally constrained by rendering quality and the physical realism of the simulator. Visual artifacts, simplified contact dynamics, and inaccurate physical modeling may introduce discrepancies from real-world behavior. Consequently, while synthetic data is significantly more scalable and cost-effective, it often exhibits lower realism compared to real-world datasets.

A common strategy for synthetic data generation is procedural randomization within simulation environments. The GraspVLA work (Deng et al., 2025) introduces the large-scale synthetic grasp dataset SynGrasp-1B, which leverages extensive variation in object appearance, scene parameters, and viewpoints to encourage the learning of robust geometric features, particularly for quasi-static grasping tasks. Although primarily simulator-driven, GraspVLA incorporates real-world data to mitigate the sim-to-real gap and improve deployment performance. Similarly, simulator-based corpora such as RoboCasa (Nasiriany et al., 2024) scale household manipulation by providing diverse kitchen environments, asset libraries, and structured task suites, enabling large-scale synthetic rollouts and demonstration collection for training generalist policies.

Beyond randomization, more automated simulation pipelines have emerged. Frameworks such as RoboGen (Wang et al., 2024c) employ large language models to propose tasks and automatically generate simulation code to expand task diversity; validation and filtering mechanisms are typically required to remove noisy or physically implausible generations. Demonstration augmentation methods such as MimicGen (Mandlekar et al., 2023) further scale simulator data by perturbing object poses and initial conditions from a small set of human seed demonstrations, increasing dataset size while preserving underlying task structure.

Synthetic data is frequently used to bootstrap policies before subsequent calibration with real-world data. Large-scale procedural corpora such as SynGrasp-1B, introduced in GraspVLA (Deng et al., 2025), provide extensive randomized scenes to support robust geometric pretraining, while simulator-based environment suites such as RoboCasa (Nasiriany et al., 2024) enable scalable household manipulation rollouts for imitation learning and generalist policy pretraining. More automated pipelines, including task-generation frameworks like RoboGen (Wang et al., 2024c), expand task diversity with reduced manual authoring, and augmentation methods such as MimicGen (Mandlekar et al., 2023) scale limited human demonstrations within simulation by perturbing object configurations and initial conditions. Although we categorize them by primary role, in practice many resources serve dual purposes. For instance, LIBERO (Liu et al., 2023), CALVIN (Mees et al., 2022), Meta-World (Yu et al., 2021), RLBench (James et al., 2020), BEHAVIOR-1K (Li et al., 2024a), and VLABench (Zhang et al., 2024b) are used as VLA fine-tuning datasets. We will discuss these works in detail in the benchmark and data engine sections.

Despite their scalability, synthetic datasets remain limited in fidelity. Simulated imagery often diverges from real-world observations due to rendering artifacts and simplified scene modeling, while physical dynamics such as contact and friction are difficult to reproduce accurately. In grasping tasks, poses are frequently generated or filtered through heuristic sampling, which may not reflect stable or efficient manipulation strategies. As a result, synthetic data is typically used for large-scale pretraining or augmentation, with real-world data required for final calibration and deployment.

## 4  VLA Benchmarks

A VLA benchmark is an evaluation dataset designed to assess the performance and generalization ability of a VLA model deployed on a robot. Compared with training datasets, benchmarks are typically smaller in scale but constructed with representative tasks and well-defined evaluation metrics to enable standardized comparison. Since real-robot evaluation is costly and operationally complex, most benchmarks are implemented in simulation environments. To systematically characterize existing VLA benchmarks, we analyze them along two analytical dimensions, namely task complexity and environment structure, as illustrated in Figure 3. Task complexity reflects the compositional and temporal difficulty of manipulation objectives, whereas environment structure captures scene diversity and spatial variability. In many benchmarks, these two factors vary simultaneously, entangling multiple sources of difficulty within a single evaluation setting. Given that VLA models tightly couple perception, language grounding, and control, such en-

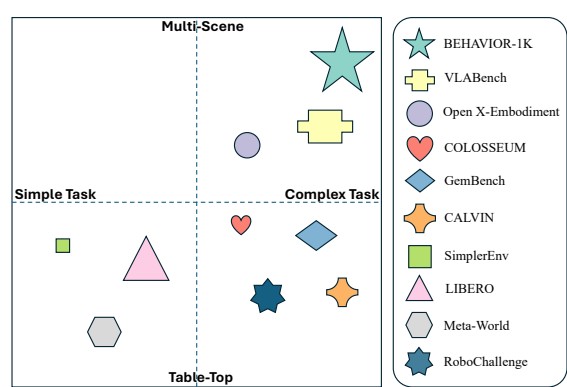

Figure 3: Task–environment landscape of VLA benchmarks. Benchmarks are positioned according to task complexity (horizontal axis) and environment structure (vertical axis). Marker size roughly reflects relative task or dataset scale.

tanglement makes failure attribution challenging. By organizing benchmarks within this two-dimensional landscape, we provide a more interpretable perspective on the aspects of VLA capability that each benchmark emphasizes. Although benchmarks are primarily designed for evaluation, many of them provide explicit training and testing splits due to the difficulty of achieving true generalization in VLA systems. Such designs enable controlled assessment of a model's learning and transfer capabilities under standardized protocols. Simulator-based suites such as LIBERO (Liu et al., 2023) and Meta-World (Yu et al., 2021), for example, are frequently used in both training and evaluation. Accordingly, we discuss benchmarks not only as evaluation protocols but also, when relevant, as structured data resources that support model development.

### 4.1  Table-top Benchmarks

Table-top benchmarks evaluate VLA models under constrained table-top tasks, which are the most common tasks that are clean enough. Researchers are also easy to reproduce the table-top evaluation results in real-

world experiments. Existing table-top benchmarks can be broadly divided into simple short-horizon tasks and complex long-horizon compositional tasks.

The simple table-top category includes benchmarks that focus on atomic manipulation tasks executed within short action horizons under constrained table-top environments. Meta-World (Yu et al., 2021), for example, comprises 50 simple manipulation tasks within a shared table-top setting, and often relies on low-dimensional state observations, which substantially simplify visual perception and scene understanding. LIBERO (Liu et al., 2023) follows a similar short-horizon design, where most tasks correspond to atomic skills that can be completed within limited steps; although procedural variations in object types and spatial arrangements are introduced, interaction remains confined to table-top settings and long-range temporal dependencies are only weakly represented. There are also a series of works built on LIBERO, named LIBERO-plus (Fei et al., 2025), LIBERO-PRO (Zhou et al., 2025), LIBERO-X Wang et al. (2026) that enhance the robustness and fairness of the benchmark. SimplerEnv (Li et al., 2024b) evaluates policies on short-horizon table-top manipulation tasks and deliberately maintains environments that are only sufficiently realistic to preserve sim-to-real ranking consistency, prioritizing execution reliability over compositional reasoning. Overall, these benchmarks emphasize immediate action correctness and low-level control stability, while placing limited stress on long-horizon reasoning or complex environmental variation. Beyond simulation platforms, Yakefu et al. (2025) provide a real-world table-top robotic platform that enables online evaluation of VLA models.

Another line of work constructs complex, long-horizon VLA benchmarks within constrained table-top environments, generating compositional tasks while maintaining a simplified interaction setting. CALVIN evaluates long-horizon manipulation by requiring agents to execute extended sequences of unconstrained language instructions across multiple tabletop environments, with the most challenging protocol requiring zero-shot generalization to an unseen environment, thereby isolating sustained grounding and temporal credit assignment as primary challenges (Mees et al., 2022). By deliberately limiting perceptual and environmental variability, it exposes rapid performance degradation as instruction chains lengthen. GemBench extends this perspective by systematically assessing hierarchical generalization across novel object placements, unseen instances, and compositional long-horizon tasks within the RLBench simulator, revealing severe performance bottlenecks at higher complexity levels despite strong short-horizon results (Garcia et al., 2025). COLOS-SEUM further evaluates robustness under controlled table-top settings by introducing systematic visual and physical perturbations across 14 axes, demonstrating significant degradation when multiple perturbation factors are applied simultaneously(Pumacay et al., 2024). Overall, these benchmarks emphasize reasoning difficulty, compositional grounding, and robustness under controlled environmental conditions rather than expanding environment scale or visual diversity.

## 4.2 Multi-scene Benchmarks

Multi-scene benchmarks aim to evaluate embodied agents under substantially more complex task and environment conditions. In contrast to table-top settings, these benchmarks emphasize interaction across diverse scenes, long-horizon execution, and compositional reasoning, reflecting a shift toward more realistic and semantically rich embodied tasks.

Complex multi-scene benchmarks typically involve long-horizon tasks executed across diverse environments, and most existing benchmarks in this category focus on extended episodes with substantial compositional complexity. BEHAVIOR-1K evaluates everyday human activities that unfold over long durations and require coordination of multiple manipulation skills, with tasks specified in a predicate-based language that explicitly encodes multi-stage objectives for structured long-horizon assessment (Li et al., 2024a). The benchmark spans full-room and multi-room environments and supports realistic physical interactions involving rigid objects, deformable materials, and fluids. VLABench further increases task complexity by constructing composite language-conditioned tasks that integrate multiple skills with long-horizon multi-step reasoning and intermediate reasoning grounded in scene semantics (Zhang et al., 2024b). It introduces substantial environmental diversity through varied scene types, object categories, and randomized configurations, thereby amplifying both perceptual and structural difficulty. Open X-Embodiment adopts a complementary scale-driven perspective by aggregating data from heterogeneous real-world robots and environments, emphasizing behavioral breadth and cross-embodiment transfer rather than enforcing a unified task structure or explicitly designed long-horizon objectives (O'Neill et al., 2025). Across these benchmarks, difficulty arises from the

Table 2: Overview of representative VLA data engines. Each engine is summarized by its input source, whether it relies on real robot hardware (automation level), requires human operation or annotation (deployment cost), includes sim-to-real validation (practicality), and its core technical contributions

| Engine | Input | Real Robot | Human | Sim2Real | Key Feature |
|---|---|:---:|:---:|:---:|---|
| *Video-to-Data Engines* | | | | | |
| **H2R** | Egocentric video | ✗ | ✗ | ✓ | Hand-to-robot retargeting via inpainting |
| **RoboWheel** | Egocentric video | ✗ | ✗ | ✓ | Physics-aware SDF + residual RL retargeting |
| **Video2Policy** | Internet video | ✗ | ✗ | ✓ | GPT-4o executable task code generation |
| **X-Humanoid** | Ego-Exo4D video | ✗ | ✗ | ✓ | Humanoid-specific video diffusion robotization |
| **GenMimic** | Video generation output | ✗ | ✗ | ✓ | Zero-shot transfer from video gen models |
| **UniSim** | Internet video + robot data | ✗ | ✗ | ✓ | Conditional video diffusion world model |
| *Hardware-Assisted Engines* | | | | | |
| **ALOHA** | Bimanual teleoperation | ✓ | ✓ | ✗ | Kinematic isomorphism, $20k hardware |
| **GELLO** | Teleoperation | ✓ | ✓ | ✗ | $300 3D-printed exoskeleton, 30% reliability gain |
| **UMI** | In-the-wild demo | ✓ | ✓ | ✗ | GoPro gripper + SLAM, 3× faster collection |
| **DexCap** | In-the-wild demo | ✓ | ✓ | ✗ | EMF gloves + RGB-D dexterous manipulation |
| **Lucid-XR** | VR + simulation | ✗ | ✓ | ✓ | Physics sim on VR headset + diffusion rendering |
| *Generative Data Engines* | | | | | |
| **MimicGen** | Simulation | ✗ | ✗ | ✗ | Object-centric subtask trajectory reuse |
| **DynaMimicGen** | Simulation | ✗ | ✗ | ✗ | DMP adaptation for moving objects |
| **DemoGen** | 3D point cloud | ✗ | ✗ | ✗ | Fully synthetic, no real robot needed |
| **GenSim** | LLM + simulation | ✗ | ✗ | ✗ | LLM-generated task code and scene configs |
| **RoboGen** | LLM + simulation | ✗ | ✗ | ✗ | RL + motion planning per subtask |
| **RoboTwin 2.0** | LLM + simulation | ✗ | ✗ | ✗ | VLM feedback loop + domain randomization |
| **ROSIE** | Real demo + diffusion | ✓ | ✓ | ✗ | Text-to-image semantic inpainting |
| **RoboEngine** | Real demo + diffusion | ✓ | ✓ | ✗ | Plug-and-play Robo-SAM augmentation |
| **EMMA** | Real demo + diffusion | ✓ | ✓ | ✗ | Multi-view consistent DreamTransfer |
| **PointWorld** | World model | ✗ | ✗ | ✓ | 3D point flow for zero-shot MPC |
| **IRASim** | World model | ✗ | ✗ | ✓ | Trajectory-to-video diffusion evaluation |
| **3D-VLA** | World model | ✗ | ✗ | ✓ | 3D multimodal goal state generation |
| **Genie** | Internet video | ✗ | ✗ | ✗ | Unsupervised latent action discovery |

joint expansion of task horizon and environmental variability, stressing compositional reasoning, robustness, and generalization across scenes and embodiments.

# 5 VLA Data Engines

A VLA data engine refers to a scalable system or pipeline designed to continuously generate, transform, or augment training data for VLA models. Unlike datasets, which are typically collected once and reused as static corpora, data engines emphasize the algorithm to generate high-quality datasets.

Formally, a data engine can be viewed as a data generation process that produces robotics data required by VLA training and evaluation. Depending on the level of automation and physical grounding, data engines rely on video reconstruction, hardware-assisted teleoperation, generative simulation, or learned world models. By shifting from static collection to dynamic generation, data engines aim to address the scalability limitations of real-world data acquisition while maintaining sufficient task structure and embodiment alignment.

## 5.1 Video-to-Data Engine

Video-to-data engines transform human demonstration videos into robot-executable training data, addressing VLA's data scarcity by leveraging web-scale video resources. The core challenge lies in bridging the visual embodiment gap: human hands and bodies differ fundamentally from robot manipulators in both appearance and kinematics, hindering direct policy transfer.

A series of work detects and reconstructs the essential poses and actions in the video with optimizations related to VLA data construction. Egocentric video editing directly replaces human hands with rendered robot arms. H2R (Li et al., 2026) detects 3D hand poses, retargets motions to robot kinematics, and composites robot arms onto videos using segmentation and inpainting, improving manipulation success by 1.3%–10.2% in simulation and 3%–23% on real robots when pretraining visual encoders (MAE (He

et al., 2022), R3M (Nair et al., 2022)). RoboWheel (Zhang et al., 2025b) extends this with physics-aware optimization through SDF penalties and residual RL, ensuring contact timing and grasp semantics are preserved while supporting cross-embodiment retargeting to 6/7-DOF arms, dexterous hands, or humanoid robots. These approaches preserve scene context and dynamics while explicitly bridging the visual domain gap, critical for VLA systems where language instruction grounding relies on precise manipulation primitives that must maintain consistent semantics across different robot embodiments.

Scene reconstruction methods extract structured task representations rather than preserving raw pixels. Video2Policy (Ye et al., 2025) extracts object meshes and 6D poses, then uses GPT-4o to generate executable task code with iterative refinement, achieving 88% simulation success rate across 100+ videos and enabling sim-to-real VLA transfer with clearer task structure and automated language annotations. For whole-body humanoid robots, X-Humanoid (Yang et al., 2025) fine-tunes video diffusion models (e.g., Wan; Wan et al., 2025) to "robotize" entire human bodies, converting 60+ hours of Ego-Exo4D (Grauman et al., 2024) into humanoid demonstrations while preserving full-body dynamics. GenMimic (Ni et al., 2025) learns directly from video generation model outputs, lifting synthetic human motions to 4D with weighted keypoint tracking and symmetry regularization, achieving zero-shot transfer to physical robots and suggesting future VLA systems could train on text-to-video outputs without real human demonstrations. UniSim (Yang et al., 2024) represents the most general approach, learning a conditional video diffusion model from internet images/videos and robot data to enable autoregressive simulation of long-horizon interactions. UniSim allows VLA policies to train in closed-loop, achieving 3–4× better performance than short-demonstration baselines with zero-shot real-robot transfer. Since even tiny error can lead to unstable VLA training, these data engines still face challenges in reducing reconstruction noise and acquiring minimal action errors.

## 5.2   Hardware-Assisted Engine

Hardware-assisted engines collect VLA data by controlling the robot action by the action sensor in hardware, enabling real-time action capture without complex 3D reconstruction. The core challenge lies in balancing cost-effectiveness, ergonomic design, and capturing sufficient signals for VLA training.

Robot-to-robot teleoperation provides intuitive control through kinematic isomorphism. ALOHA (Zhao et al., 2023) achieves fine-grained bimanual tasks on hardware less than 20k dollars cost, reaching 80-90 percents success rate when combined with ACT's action chunking, while GELLO (Wu et al., 2024) further reduces cost to less than 300 dollars through 3D-printed exoskeletons with passive joint regularization, improving reliability by almost 30 percents over VR baselines. However, lab-based setups limit scene diversity.

Portable interfaces address this by trading precision for scalability. UMI (Chi et al., 2024) uses a GoPro-equipped gripper with SLAM tracking to collect demonstrations across 30 real-world locations in 12 person-hours, which is 3× faster than standard teleoperation while achieving 71.7 percents zero-shot success. Dex-Cap (Wang et al., 2024a) targets dexterous manipulation through EMF gloves and chest-mounted RGB-D cameras, achieving 72 percents success on multi-finger tasks via IK retargeting and point cloud-based policies. Both systems enable in-the-wild data collection while maintaining cross-embodiment transfer capabilities critical for VLA deployment on heterogeneous hardware.

XR-augmented simulation merges hardware assistance with synthetic generation. Lucid-XR (Ravan et al., 2025) runs physics simulation directly on VR headsets at <12ms latency, then applies diffusion models to transform rendered observations into photorealistic images, achieving 5× effective data compared to real teleoperation with superior robustness to environment changes.

As costs decrease and XR hardware improves, hybrid approaches combining teleoperation precision, portable scalability, and generative augmentation will likely dominate VLA training pipelines.

## 5.3   Generative Data Engine

Generative engines address VLA's data scarcity through scalable synthetic data generation and visual augmentation to create diverse training datasets without physical robot deployment. The core challenge lies in minimizing human intervention, covering diverse task and scene, and transferability.

The most established approach employs trajectory reuse through task decomposition. MimicGen (Mandlekar et al., 2023) pioneered this paradigm by segmenting demonstrations into object-centric subtasks, then spatially transforming these segments to new object configurations. It generates 50k demonstrations from only 200 human seeds across long-horizon assembly tasks. DynaMimicGen (Pomponi et al., 2025) extends this with Dynamic Movement Primitives, enabling real-time adaptation to moving objects during trajectory generation, critical for dynamic tasks where static assumptions fail. DemoGen (Xue et al., 2025) eliminates the need of real robots through fully synthetic 3D point cloud editing: it segments, transforms, and composites object point clouds to generate both actions and observations, achieving 74.6 percents average success across eight real-world tasks from single demonstrations. For VLA training, these methods mainly serve as data augmentation engines that amplify limited human supervision into large-scale datasets.

LLM-driven data generation automates the creation of entirely new tasks and environments. GenSim (Wang et al., 2024b) and RoboGen (Wang et al., 2024c) query LLMs to generate simulation task code, scene configurations, and reward functions, bootstrapping diverse task libraries (100+ tasks) from minimal human prompts. RoboGen further integrates multiple learning algorithms (RL, motion planning, trajectory optimization) selected per-subtask, achieving 77.4% average success across 69 benchmark tasks. RoboTwin 2.0 (Chen et al., 2025) enhances this with multimodal LLM feedback loops: a VLM observer monitors simulation execution, detects failures, and provides corrections to iteratively refine task code, improving success rates while reducing token costs. Combined with comprehensive domain randomization (scene clutter, lighting, textures, table heights, diverse language instructions), RoboTwin 2.0 generates 100k+ expert trajectories across five robot platforms. For VLA systems, these LLM-driven engines work well at pre-training stage. They can expand task coverage before real-data finetuning and provide benchmark-aligned evaluation automatically, though their performance is limited by LLM performance.

Visual augmentation through generative models transforms limited real demonstrations into visually diverse datasets. ROSIE (Yu et al., 2023)applies text-to-image diffusion for semantic inpainting, replacing objects and backgrounds in robot demos to create unseen tasks (e.g., swapping chip bags for towels), improving overall performance by over 115%. RoboEngine (Yuan et al., 2025) packages this into a plug-and-play toolkit with Robo-SAM (a robot-specific segmentation model trained on the new RoboSeg dataset) and physics-aware background generation, achieving similar improvements without prerequisites like green screens or camera calibration. EMMA (Dong et al., 2025) targets multi-view consistency through DreamTransfer, a diffusion transformer that generates geometrically coherent videos across camera angles while enabling text-controlled editing of foregrounds/backgrounds/lighting.

Predictive world models enable closed-loop training by forecasting environment responses to actions. Point-World (Huang et al., 2026) represents states and actions as 3D point flows for geometric precision, enabling zero-shot MPC deployment, though it lacks visual textures for vision-based policies. IRASim (Zhu et al., 2025) addresses this through trajectory-to-video diffusion with frame-level action conditioning, achieving 0.99 correlation with ground-truth simulation for evaluation and improving Push-T performance from 0.637 to 0.961 IoU via planning, demonstrating that learned visual dynamics can replace expensive real-robot testing. 3D-VLA (Zhen et al., 2024)bridges geometric and visual prediction by generating multimodal goal states (RGB, depth, point clouds) through diffusion models aligned with 3D-LLM, showing that imagining future 3D states improves VLA action planning. Genie (Bruce et al., 2024) explores unsupervised learning from 200k hours of internet videos, discovering latent actions through VQ-VAE without robot labels, suggesting VLA systems could bootstrap world models from web-scale data, though 1 fps generation and 16-frame memory limit real-time deployment. These approaches offer complementary VLA capabilities: geometric planning (PointWorld), efficient evaluation (IRASim), goal-conditioned supervision (3D-VLA), and unsupervised action discovery (Genie).

Generative engines employ diverse automation strategies. Trajectory reuse (MimicGen family) maximizes demonstration efficiency but assumes known subtask structures; LLM-driven generation (GenSim/RoboGen/RoboTwin) automates task creation but remains simulation-bound; visual augmentation (ROSIE/RoboEngine/EMMA) enriches real data but cannot generate new physics; predictive world models (PointWorld/IRASim/3D-VLA/Genie) enable interactive training, efficient evaluation, and goal-conditioned planning, though they require either massive pretraining on internet videos or careful alignment between predicted and real dynamics. As LLM grounding improves and video generation advances, hybrid pipelines

combining task generation, trajectory synthesis, visual augmentation, and predictive modeling will likely become standard VLA pretraining infrastructure, providing both the task diversity needed for generalist capabilities and the data scale required for robust real-world deployment.

## 6 Limitations, Open Problems, and Future Directions

### 6.1 Dataset Limitations: Fidelity-Cost Trade-off

A central limitation of existing VLA datasets lies in the trade-off between data fidelity and scalability. High-fidelity real-world datasets provide accurate visual observations and physically grounded trajectories, but they are expensive to collect and difficult to scale across tasks and robots. Aggregated corpora such as Open X-Embodiment increase trajectory volume and robot diversity (Padalkar et al., 2023; O'Neill et al., 2025), yet such scale is achieved by combining heterogeneous interfaces and action parameterizations, which introduces alignment complexity. In contrast, single-platform datasets such as RT-1, DROID, and BridgeData V2 offer greater interface consistency and controlled data collection (Brohan et al., 2022; Khazatsky et al., 2024; Walke et al., 2023), but their embodiment scope and task diversity are comparatively limited. As a result, improving scale often comes at the expense of interface consistency, while maintaining high fidelity and standardized control restricts diversity and expansion.

This trade-off is further amplified in multimodal and semantic supervision. Contact-rich behaviors requiring force or tactile feedback remain underrepresented because collecting such signals in real environments is costly and hardware-dependent. Although multimodal datasets such as RH20T incorporate additional sensing modalities (Fang et al., 2023), their scale remains small compared with vision-language data. Similarly, semantic expansion through large human-centric datasets or co-training pipelines (Brohan et al., 2023; Grauman et al., 2022; Zhang et al., 2025c) increases linguistic diversity at relatively low cost, yet grounding these semantics into physically valid closed-loop robot control requires expensive real-world interaction data. Overall, current dataset development strategies have not resolved the fundamental tension between fidelity and cost, and scaling high-quality embodied supervision remains a core challenge for VLA research.

### 6.2 Benchmark Limitations: Lacking Benchmarks for Reasoning Ability in VLA

Current VLA benchmarks increasingly reveal weaknesses in temporal and compositional reasoning, yet few are explicitly designed to diagnose these capabilities in a structured manner. Performance degradation in long-horizon tasks often reflects more than cumulative control error; it exposes limitations in temporal abstraction, memory retention, and multi-step planning. For example, in CALVIN, success rates drop to 0.08 percent for five sequential instructions (Mees et al., 2022), while VLABench reports systematic failures in multi-step logical tasks (Zhang et al., 2024b). However, such benchmarks primarily measure overall success without disentangling whether failures arise from deficient planning, unstable memory, poor skill composition, or inadequate recovery mechanisms. As a result, they expose symptoms of reasoning failure but provide limited diagnostic structure.

A similar limitation appears in generalization evaluation. Many benchmarks vary individual factors such as object identity or scene layout in isolation, whereas real-world deployment requires robustness under compounded variability across perception, embodiment, and semantics. THE COLOSSEUM demonstrates substantial performance degradation under combined perturbations (Pumacay et al., 2024), suggesting that single-axis robustness does not extrapolate to multi-axis settings. Although cross-embodiment datasets broaden platform diversity (O'Neill et al., 2025), evaluation protocols often remain task-specific and horizon-limited. Overall, current benchmarks lack structured frameworks for assessing reasoning ability in VLA systems, particularly in settings where temporal composition, multi-factor variability, and cross-embodiment transfer must be jointly evaluated.

### 6.3 Data Engine Limitations: Scaling Generation Without Scaling Grounding

The primary limitation of current data engines is not generation capacity but grounding reliability. Video-based pipelines depend critically on perception fidelity. Failures in grounding, pose estimation, and depth

reconstruction introduce systematic noise that propagates into policy learning (Ye et al., 2025; Ni et al., 2025; Zhang et al., 2025b; Xue et al., 2025). Even when trajectories can be synthesized at scale, physical plausibility is not guaranteed. Editing and interpolation methods require feasibility checks or assume structured subtasks (Li et al., 2026; Mandlekar et al., 2023; Pomponi et al., 2025), while hardware systems face calibration and embodiment constraints (Zhao et al., 2023; Wang et al., 2024a; Chi et al., 2024). LLM-driven engines further reveal gaps in physical understanding and reward specification (Wang et al., 2024b;c).

Interactive world models offer a more unified approach, yet remain constrained by limited temporal context, computational cost, and sim-to-real discrepancies (Yang et al., 2024; Bruce et al., 2024; Zhu et al., 2025; Zhen et al., 2024; Chen et al., 2025). The common thread across these engines is a scaling imbalance: data generation scales faster than physical grounding, verification, and embodiment alignment. Future progress therefore requires integrating physics constraints, temporally coherent reconstruction, and embodiment-aware reasoning into generative pipelines. Rather than treating video, simulation, and hardware capture as separate paradigms, a promising direction is the development of unified engines that couple semantic generation with physically validated control, ensuring that scalability does not outpace reliability.

### 6.4 Future Directions

Looking forward, we argue that scalable synthetic data generation will play an increasingly central role in VLA development. However, the primary obstacle is no longer data scale, but the fidelity gap between synthetic environments and real-world deployment settings. Given that VLA systems are highly sensitive to the specific scenes, future progress will likely depend on systematically bridging this sim-to-real quality gap rather than merely expanding synthetic diversity. A promising direction is to reconstruct real-world scenes inside simulation environments with high geometric and physical fidelity. Instead of relying on abstract procedural layouts, future data engines may aim to faithfully digitize real manipulation spaces and integrate them with physically accurate simulators. In such settings, robot planning algorithms could automatically generate large volumes of task-consistent trajectories while preserving real-world constraints. Crucially, evaluation benchmarks should also be built on top of these high-fidelity data engines to ensure that training and testing remain aligned with practical deployment conditions. In the near term, integrating 3D sensing pipelines with robot platforms to construct accurate scene models offers a practical pathway toward this vision. In the longer term, advances in learned world models may enable automatic reconstruction and simulation of realistic environments from limited observations, reducing human effort while maintaining physical plausibility. Bridging synthetic scalability with real-world fidelity through next-generation data engines may therefore become a foundational step toward general intelligence of VLA models.

## 7    Conclusion

VLA research is fundamentally shaped by the data and evaluation infrastructures that support it. In this survey, we provided a structured, data-centric perspective by organizing existing resources into three complementary categories: datasets, benchmarks, and data engines. Rather than treating these as isolated components, we analyzed how they jointly determine representation learning, capability measurement, and scalability. Across datasets, we identified a persistent tension between embodiment diversity and interface consistency, revealing that scaling data volume alone does not guarantee representational alignment or generalization. In benchmarks, we highlighted structural limitations in current evaluation protocols, where long-horizon reasoning, compositional generalization, and robustness under compounded variability remain insufficiently disentangled. In data engines, we observed that generation capacity is advancing rapidly, yet physical grounding, embodiment alignment, and reliability verification lag behind scalability.

Taken together, these findings suggest that the central challenge of VLA is not merely data scarcity, but the lack of unified abstractions that bridge perception, language grounding, and embodied control across heterogeneous platforms. Future progress will likely depend on co-designing datasets, benchmarks, and generative engines under shared structural principles, enabling scalable yet physically grounded supervision. We hope this survey clarifies the current landscape and provides a foundation for building more robust, interpretable, and generalizable VLA systems.

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
