# OpenReview forum: "Vision-Language-Action in Robotics: A Survey of Datasets, Benchmarks, and Data Engines"
_TMLR — Accepted by TMLR_

### Review · Reviewer_fSZi · 2026-03-15

**Summary Of Contributions:**

This paper presents a systematic, data-centric survey of Vision-Language-Action (VLA) models in robotics. Instead of focusing purely on model architectures, the authors rightly emphasize the underlying data infrastructure, organizing the literature into three primary pillars: datasets, benchmarks, and data engines. The paper categorizes existing resources, highlights the fidelity-cost trade-off in datasets , identifies structural gaps in evaluating compositional generalization , and discusses the limitations of current generative data engines in sim-to-real transfer.

The three-way taxonomy (datasets, benchmarks, data engines) is highly effective and provides a fresh perspective on the bottlenecks in embodied AI.

The temporal landscape diagram (Figure 1) is a very helpful visual aid for mapping recent progress.

The discussion on the gap between scalable generation and physical grounding is insightful.

But,

Minor formatting and typographical inconsistencies in the references section.

The discussion on how VLA models handle out-of-distribution (OOD) scenarios and cross-domain shifts could be slightly expanded to make the survey more comprehensive.

**Audience:**

Yes

**Audience Explanation:**

The intersection of large vision-language models and embodied control is currently one of the most active research areas in machine learning. TMLR readers working on robotics, imitation learning, and multimodal foundation models will find this consolidated overview of data resources and evaluation protocols extremely useful for benchmarking their own methods.

**Broader Impact Concerns:**

No major ethical concerns are introduced directly by this survey. The authors adequately review the landscape. However, it is worth noting that scaling VLA models using internet-scraped human videos (as discussed in Section 5.1) inherently carries risks related to human bias and privacy. A brief sentence acknowledging this in the conclusion or limitations section would suffice.

**Claims And Evidence:**

Yes

**Claims Explanation:**

The submission is a survey paper, and its primary claims are organizational and analytical. The authors support their taxonomy by extensively citing and categorizing state-of-the-art works from 2023 to 2025. Their conclusions regarding the fidelity-cost trade-off and the lack of reasoning benchmarks are well-supported by their analysis of specific datasets (e.g., Open X-Embodiment, RT-1) and benchmarks (e.g., CALVIN, COLOSSEUM).

**Requested Changes:**

To strengthen the work and secure my recommendation for acceptance, I suggest the following minor adjustments:

    Expand on Cross-Domain and OOD Challenges (Strengthen the work): While the paper excellently discusses the "sim-to-real gap" and the "fidelity-cost trade-off", VLA models also suffer heavily from hallucination and domain shift when moving between vastly different environments or embodiments. It would be highly beneficial to briefly touch upon advanced domain adaptation and hallucination detection techniques. Specifically, I recommend the authors discuss the theoretical underpinnings of domain shift by referencing "Open set domain adaptation: Theoretical bound and algorithm", and address how we might detect VLA reasoning failures across domains by discussing "Beyond In-Domain Detection: SpikeScore for Cross-Domain Hallucination Detection". Adding a short paragraph integrating these concepts into Section 6 (Limitations) would make the future directions more robust.

    Reference Formatting (Critical for final camera-ready): Please do a careful pass of the bibliography. For example, some references have incomplete venue information or inconsistent capitalization (e.g., "Rt-1" vs "RT-1", and some arXiv preprints could be updated to their published versions if available).

    Table 1 and 2 Clarity: In Table 1, ensure that the checkmarks and 'X' marks are perfectly aligned; there is a slight visual artifact in the "Img/Vid/Txt" columns for some rows (e.g., SynGrasp-1B shows "✔X" which is confusing).

---

> ### Author Response · Authors · 2026-03-20
>
> **Response to Reviewer**
>
> Thank you for the detailed and constructive feedback. The paper will be revised accordingly to improve clarity and coverage.
>
> **OOD and cross-domain challenges**
>
> This is a helpful suggestion. We agree that cross-domain generalization and hallucination-related failures are also central challenges for VLA systems. We will expand the discussion in the limitations section to more explicitly cover OOD behavior, including issues such as domain shift across embodiments and environments, as well as potential failure modes. We will also incorporate relevant connections to domain adaptation and failure detection where appropriate, to make this part more complete.
>
> In our understanding, hallucination in VLA models differs from that in language models. In VLA systems, hallucinations (e.g., attempting to pick objects that are not present, or repeatedly selecting a specific object regardless of the prompt) are often caused by insufficient model training or limitations in the dataset. This suggests that the issue is closely tied to data quality and coverage, and highlights the ongoing challenge of scaling datasets and improving training strategies for general VLA models. We agree that this is an interesting point and is valuable to discuss in the revised paper.
>
> **Reference formatting**
>
> We will perform a careful pass over the bibliography to ensure consistency in formatting. This includes standardizing capitalization, completing venue information where available, and updating arXiv entries to published versions when applicable.
>
> **On table formatting**
>
> Thanks for pointing this out. We carefully checked Table 1 in the submitted version and did not observe misalignment on our side, so this may be due to rendering differences across viewers. We will double-check the formatting and ensure the table is cleanly aligned in the final version.
>
> **Broader impact**
>
> We will add a brief note discussing potential risks associated with large-scale data collection, including bias and privacy considerations when using internet-scale human data.

---

### Review · Reviewer_Qnf9 · 2026-03-16

**Summary Of Contributions:**

This paper presents a data-centric survey of  VLA research in robotics, focusing on three key components: datasets, benchmarks, and data engines. The survey organizes existing work into a unified taxonomy and analyzes how these components jointly influence the development and evaluation of VLA systems. In addition, the paper proposes a structured analytical framework for benchmarks based on task complexity and environment structure, and identifies several open challenges such as representation alignment, long-horizon reasoning evaluation, and scalable data generation with physical grounding.

Strengths:
The paper provides a clear and well-structured taxonomy that organizes the rapidly growing VLA literature.
The data-centric perspective is timely and valuable, as most prior surveys focus primarily on model architectures.
The benchmark analysis framework offers a useful lens for understanding evaluation gaps in current VLA research.
The survey covers a wide range of representative datasets, benchmarks, and data engines and provides consistent comparisons across them.

Weaknesses:
Some sections (e.g., hardware-assisted engines) are more descriptive than analytical.
The survey could benefit from additional quantitative comparisons across benchmarks or datasets.

**Additional Comments:**

Overall, this is a well-written and timely survey that provides a clear overview of the emerging VLA research landscape from a data-centric perspective. The proposed taxonomy and benchmark analysis framework are useful conceptual tools that can help researchers better understand existing resources and identify open challenges in the field. With minor clarifications and improvements, the paper will serve as a valuable reference for the community.

**Audience:**

Yes

**Audience Explanation:**

VLA models represent a rapidly developing area at the intersection of machine learning, robotics, and embodied AI. Researchers working on representation learning, embodied reasoning, robot learning, and evaluation methodology would likely find this survey useful. By organizing the literature around datasets, benchmarks, and data engines, the paper provides a helpful overview of the current research landscape and highlights important open problems that may guide future work.

**Broader Impact Concerns:**

This work is a survey paper and does not introduce new models or systems. As such, it does not raise significant ethical concerns. However, the authors may optionally include a brief discussion of the broader implications of large-scale robotic data collection and deployment of VLA systems in real-world environments.

**Claims And Evidence:**

Yes

**Claims Explanation:**

The claims in the paper are generally well supported by references to representative datasets, benchmarks, and data engines in the VLA literature. The authors systematically analyze these resources and provide concrete examples to illustrate key observations such as the fidelity–cost trade-off in datasets, the limitations of current benchmarks in evaluating long-horizon reasoning, and the scalability challenges of current data engines.

The discussion is consistent with the cited literature and the conclusions are appropriate in scope for a survey paper.

**Requested Changes:**

The authors could briefly clarify the criteria used to select representative works included in the survey.

Some sections (particularly in the data engine discussion) could benefit from more analytical comparisons rather than descriptive summaries.

A small summary table comparing benchmarks (e.g., number of tasks, horizon length, or evaluation metrics) could further strengthen the benchmark analysis.

---

> ### Author Response · Authors · 2026-03-20
>
> **Response to Reviewer**
>
> Thank you for the helpful suggestions. We will revise the paper accordingly to improve clarity and strengthen the analytical components.
>
> **Paper selection criteria**
>
> We agree with the reviewer that the current draft does not clearly state how representative works are selected. We will make this explicit in the introduction. In particular, the selection is guided by relevance to the data-centric perspective, coverage of key design dimensions (such as embodiment, modality, and data generation paradigms), and inclusion of widely used or representative resources in recent VLA literature. We will also briefly clarify that the goal is to provide structured coverage rather than an exhaustive list. Briefly, we select the "data centric" works in VLA rather than papers focus on algorithm or model.
>
> **Analytical depth (especially data engines)**
>
> We agree that parts of the data engine section lean more toward description. We will revise this section to make the analysis more explicit by introducing clearer comparison dimensions across approaches, including automation level, physical grounding, scalability, and embodiment alignment. The discussion will be reorganized to highlight the differences between paradigms, rather than focusing primarily on individual works.
>
> Specifically, we think different type of data engines focus on different key idea to scale the data. For example, video-to-data scale VLA data with video, since video data is sufficient, and hardware-assist data engines use human labor to scale up it. We will add a concrete discussion on the high-level ideas regarding these points.
>
> **On benchmark comparison**
>
> This is a helpful suggestion. We will add a concise summary table for benchmarks, covering key attributes such as task scale, horizon length, environment diversity, and evaluation metrics.
>
> **On clarity and consistency**
>
> We will also revise several parts of the paper to make comparisons more explicit and improve overall clarity and consistency.

---

### Review · Reviewer_MBkv · 2026-03-17

**Summary Of Contributions:**

This paper presents a data-centric survey of Vision-Language-Action (VLA) research in robotics, organized around three pillars: datasets, benchmarks, and data engines. Its main contribution is a unified taxonomy and analytical framework that clarifies how these components support training, evaluation, and scalable data generation. The paper also highlights several cross-cutting open challenges, including embodiment alignment, long-horizon reasoning evaluation, and physically grounded scalable data generation.

Strength: The paper has a clear and timely scope: focusing on the data and evaluation infrastructure behind VLA, rather than only model architectures, is valuable and well motivated. The taxonomy into datasets, benchmarks, and data engines is intuitive and useful, and the paper further refines each category in a structured way. The survey goes beyond enumeration and offers synthesis, especially in its discussion of fidelity-cost trade-offs, benchmark gaps, and grounding limitations in data engines. The paper is likely to be useful to researchers entering VLA as well as practitioners thinking about dataset design and evaluation protocols.

Weakness: Some sections remain relatively descriptive, and a few comparative claims could be sharpened further with more explicit criteria for inclusion and comparison. The paper would benefit from a slightly stronger discussion of how overlapping resources are handled, since some works function as both datasets and benchmarks or benchmarks and data engines. While the survey identifies important open problems, some recommendations for future evaluation protocols could be made more operational.

**Additional Comments:**

Overall, I am positive on this submission. I think the paper addresses an important gap, is well organized, and offers a useful framing that could help structure future VLA research. With minor revisions for clarity and positioning, I would support acceptance.

**Audience:**

Yes

**Audience Explanation:**

TMLR’s audience includes researchers interested in machine learning systems, embodied AI, robotics, evaluation, and data-centric ML. This survey addresses an active and fast-growing area, and its emphasis on datasets, benchmarks, and data engines makes it useful not only for VLA specialists but also for readers interested in general questions of scaling, evaluation, and sim-to-real transfer.

**Broader Impact Concerns:**

I do not have major broader-impact concerns beyond the standard issues associated with robotic deployment, benchmark misuse, and overinterpretation of simulation-based evaluation. A short note on these risks could further strengthen the paper, but this is not a major weakness.

**Claims And Evidence:**

Yes

**Claims Explanation:**

For a survey paper, the claims are generally appropriately supported through broad coverage, explicit categorization, and synthesis across representative works. The core claims are also modest in scope: the paper does not overclaim new empirical results, but instead argues for a data-centric framing and supports this through taxonomy, examples, and cross-sectional analysis. I found the central narrative clear and convincing overall.

**Requested Changes:**

Clarify the criteria used to select representative works, especially for rapidly evolving 2025-era papers.

Add a short discussion on category overlap and how borderline cases are assigned.

Strengthen the concluding recommendations with a more concrete checklist or set of design principles for future dataset/benchmark/data-engine development.

Minor proofreading would help in a few places to improve polish and consistency.

---

> ### Author Response · Authors · 2026-03-20
>
> **Response to Reviewer**
>
> We sincerely thank the reviewer for the careful reading of our paper and the constructive feedback. We are encouraged that the reviewer finds the paper timely, well-structured, and valuable for the community. We address the main concerns below and will incorporate the suggested revisions in the updated version.
>
> **1. Clarity of selection criteria for representative works**
>
> We appreciate the suggestion to clarify how representative works are selected, especially given the rapid evolution of VLA research.
>
> In the current version, our selection is guided by three implicit criteria:
> (i) relevance to the data-centric perspective,
> (ii) influence or adoption within the community,
> (iii) diversity in design choices (e.g., embodiment, modality, or generation paradigm).
>
> We agree that these criteria should be made explicit. In the revision, we will add a short paragraph in the introduction to clearly state the inclusion criteria to clarify how recent works are incorporated, including how we balance novelty with maturity.
>
> **2. The category overlap**
>
> The reviewer raises an important point regarding overlapping roles across categories. As also noted in the paper (e.g., some resources serve both training and evaluation purposes, this is an inherent characteristic of the VLA ecosystem.
>
> To improve clarity, we will add an explicit discussion on category boundary definitions and overlap handling, include concrete examples (e.g., LIBERO, CALVIN) to illustrate dual roles, and clarify that our taxonomy is intended as a functional abstraction rather than a strict partition.
>
> **3. Making recommendations more operational**
>
> We thank the reviewer for this valuable suggestion. Our current discussion of future directions emphasizes high-level challenges, but we agree that more actionable guidance would strengthen the paper.
>
> In the revision, we will:
> - Add a concise checklist or design principles section summarizing actionable guidelines for:
>   - dataset construction (e.g., embodiment consistency vs diversity),
>   - benchmark design (e.g., disentangling reasoning vs control),
>   - data engine development (e.g., grounding validation mechanisms),
> - Reframe parts of Section 6 into more concrete recommendations while preserving the conceptual insights.
>
> **4. Sharpening comparative claims**
>
> We appreciate the suggestion to strengthen comparative analysis. While our intent was to emphasize synthesis over exhaustive comparison, we agree that some claims can be made more precise.
>
> **5. Minor clarity, proofreading, and broader impact**
>
> We thank the reviewer for noting this. We will carefully proofread the manuscript and improve consistency in terminology and phrasing throughout. We will add a broader impact section for more explicitly discuss.

---

### Review · Reviewer_2BPr · 2026-03-21

**Summary Of Contributions:**

This paper presents a systematic, data-centric survey of Vision-Language-Action (VLA) research in robotics, organized around three pillars: datasets, benchmarks, and data engines. The authors propose a unified taxonomy that maps representative works from 2023–2025 into a tree-structured framework. For datasets, the paper categorizes real-world and synthetic corpora along embodiment diversity, modality composition, and action space formulation. For benchmarks, a two-dimensional analytical framework (task complexity × environment structure) is introduced. For data engines, existing work is classified into video-to-data, hardware-assisted, and generative paradigms. The survey distills four open challenges: representation alignment, multimodal supervision, reasoning assessment, and scalable data generation.

**Audience:**

Yes

**Audience Explanation:**

VLA is a frontier research direction in embodied AI, and data infrastructure is indeed an underappreciated yet critical bottleneck. The paper's framing—examining VLA through a data-centric lens.
Specifically, the following aspects would be of interest to the TMLR audience:
1. The three-pillar taxonomy provides a useful conceptual map for VLA researchers.
2. Tables 1 and 2 offer practical reference value for selecting training data and evaluation protocols.

**Broader Impact Concerns:**

As a survey paper, this work does not directly involve the development or deployment of new technologies, so it does not present significant ethical risks. However, the authors are encouraged to add brief discussions of the following:
1. Data privacy and consent. Some video-to-data engines leverage internet videos (including egocentric sources like Ego4D). The survey should discuss the privacy and consent implications of these data sources.
2. Bias propagation through synthetic data. LLM-driven data engines (e.g., GenSim, RoboGen) may propagate biases from language models into robot training data. This risk deserves mention.
3. Safety considerations. As VLA systems are deployed in real-world environments, data quality directly impacts operational safety. The paper should briefly discuss the safety risks arising from insufficient data quality.

**Claims And Evidence:**

No

**Claims Explanation:**

1. Figure 3's two-dimensional benchmark landscape is a valuable conceptual contribution, but the positioning of benchmarks along the axes lacks clear quantitative criteria. How is "task complexity" measured—by the number of steps, instruction length, combinatorial complexity, or something else?
2. Some discussions are overly generic. For instance, Section 6.4's discussion of "reconstructing real-world scenes inside simulation environments" does not analyze technical feasibility, the integration pathway of existing methods with VLA training pipelines, or potential technical barriers.

**Requested Changes:**

1. Formalize the benchmark analysis framework. Figure 3's two-dimensional landscape requires more rigorous definitions, e.g., explicitly define quantitative metrics for task complexity and environment structure.
2. Expand literature coverage. The survey needs more comprehensive coverage and includes a discussion of Diffusion Policy and its data efficiency properties.
3. Analyze data requirements introduced by recent VLA models.
4. Deepen the data engine comparative analysis. While Table 2 provides a useful overview, it needs additional quantitative comparison or a more specific decision tree or guideline for selecting data engines based on deployment scenarios.

---

> ### Author Response · Authors · 2026-03-23
>
> **Response to Reviewer**
>
> We thank the reviewer for the thoughtful and constructive feedback. We are encouraged that the reviewer finds the topic timely and recognizes the value of a data-centric perspective on VLA research. Below we clarify the main concerns and outline planned revisions.
>
> **Regarding the benchmark landscape** (Figure 3), we would like to clarify that our notion of task complexity is primarily grounded in whether the task requires non-trivial reasoning from the VLA model. For example, simple manipulation tasks such as pick-and-place with fixed targets fall into the low-complexity regime, while tasks that require selecting among multiple candidates based on attributes (e.g., color, size), or operating in more compositional and ambiguous environments, demand additional reasoning and are categorized as higher complexity. We agree that this distinction is currently under-specified. In the revision, we will formalize this criterion more explicitly and provide concrete examples to make the categorization clearer and more reproducible.
>
> On the comment that **some discussions are overly generic** (e.g., Section 6.4), we agree that additional technical depth would strengthen the paper. We will expand this section by outlining a more concrete pipeline for reconstructing real-world scenes into simulation and by explicitly discussing key technical challenges such as scalability, sim-to-real gaps, and integration with VLA training pipelines.
>
> Regarding **literature coverage**, we agree with the reviewer and will strengthen this aspect. In particular, we will deepen the connection between data and methods by **incorporating representative model and algorithm papers**, and explicitly analyzing what limitations they expose (e.g., data efficiency, generalization, supervision requirements). We will then discuss how these issues can be addressed from a data-centric perspective, and how existing works attempt to do so. This will better position our taxonomy as a bridge between data design and model development.
>
> For the remaining suggestions, including expanding the discussion on data requirements, improving the data engine comparison, and enriching the broader impact section, we agree and will incorporate the suggested clarifications and additions in the revision.
>
> We thank the reviewer again for the helpful suggestions and believe these revisions will improve both the clarity and practical value of the paper.

---

### Author Response · Authors · 2026-05-24

Dear Action Editor,

We recently realized that the camera-ready version we submitted may not be the correct final version of the manuscript. We sincerely apologize for the oversight.

We would like to kindly ask whether it would be possible to reopen the submission or otherwise allow us to upload the corrected camera-ready version.

We appreciate your time and consideration, and we apologize again for any inconvenience this may cause.

Best regards,
Authors

---

### Decision · Action_Editor_Hsv8 · 2026-04-24

**Recommendation:** Accept as is

**Audience:**

Yes

**Audience Explanation:**

This paper would interest TMLR readers working on embodied AI, robotics, multimodal learning, and evaluation. It offers a structured overview of VLA datasets, benchmarks, and data engines, which can help researchers understand current resources, compare design choices, and identify open challenges. Its discussion of data fidelity, scalability, benchmark design, and long-horizon evaluation is also relevant to broader machine learning audiences concerned with generalization, robustness, and reliable model evaluation.

**Claims And Evidence:**

Yes

**Claims Explanation:**

The claims are supported by accurate and convincing evidence. The paper grounds its arguments in a wide range of prior work and presents a clear taxonomy, supported by illustrative figures and detailed tables, to substantiate its analysis of datasets, benchmarks, and data engines. Key claims, such as the fidelity–cost trade-off, the limitations of synthetic data, and the challenges in evaluating long-horizon reasoning, are consistently reinforced with concrete examples from the literature. Overall, the evidence is clear, well-organized, and sufficiently comprehensive to support the main conclusions of the submission.